# NEURAL PREDICTIVE BELIEF REPRESENTATIONS

'

## ABSTRACT

Unsupervised representation learning has succeeded with excellent results in many applications. It is an especially powerful tool to learn a good representation of environments with partial or noisy observations. In partially observable domains it is important for the representation to encode a *belief state*—a sufficient statistic of the observations seen so far. In this paper, we investigate whether it is possible to learn such a belief representation using modern neural architectures. Specifically, we focus on one-step frame prediction and two variants of contrastive predictive coding (CPC) as the objective functions to learn the representations. To evaluate these learned representations, we test how well they can predict various pieces of information about the underlying state of the environment, e.g., position of the agent in a 3D maze. We show that all three methods are able to learn belief representations of the environment—they encode not only the state information, but also its uncertainty, a crucial aspect of belief states. We also find that for CPC multi-step predictions and action-conditioning are critical for accurate belief representations in visually complex environments. The ability of neural representations to capture the belief information has the potential to spur new advances for learning and planning in partially observable domains, where leveraging uncertainty is essential for optimal decision making.

## 1 INTRODUCTION

Modern supervised learning (Hastie et al., 2009a) and reinforcement learning (RL Sutton & Barto, 1998) methods have been applied successfully to many challenging applications (He et al., 2016; Sutskever et al., 2014; Mnih et al., 2015; Silver et al., 2016). On the other hand, in most domains there exists a great wealth of information in the raw data alone. Unsupervised learning provides a generic framework allowing machines to learn independently of supervision (Hastie et al., 2009b).

Unsupervised learning encompasses a wide range of learning problems (Rezende et al., 2014; Goodfellow et al., 2014; Erhan et al., 2010). Among them representation learning has drawn significant attraction in recent years (Bengio et al., 2013). In representation learning the goal of the learner is to learn a representation that encodes useful information required to solve a variety of tasks.

Representation learning is especially important in partially observable dynamical environments, such as navigation tasks with a first-person view, where each observation only provides a partial and possibly noisy view of the environment. In these settings it is critical for the agent to build a belief state representation which encodes its uncertainty about the underlying state of the environment. This is due to the fact that the belief state is a sufficient statistic for predicting future observations and future states, as well as the optimal policy in the RL setting (Rabiner, 1989; Jaakkola et al., 1995). Representation learning has been proven useful to enhance the performance of agents in various partially observable dynamical domains (Jaderberg et al., 2016; Oord et al., 2018; Eslami et al., 2018).

Despite these successes, prior work mostly evaluate the quality of the learned state representation indirectly through the performance in some supervised or RL task (Jaderberg et al., 2016). This *black-box* approach is effective for evaluating the usefulness of the learned representation for a particular task. However, it provides no answer to the question of whether the state representation encodes a belief embedding, nor whether the representation learns more general concepts that can be used across tasks.

In this paper, as an alternative to the current *black-box* approach, we adopt a *glass-box* approach to the problem of evaluating the representation learning methods. More specifically we directly use the learned representation to predict the ground-truth state of the environment. This information is only used for evaluating the representation, while the representation itself is learned in a fully unsupervised fashion.

For our experiments, we use a set of simulated tasks in the DeepMind Lab suite (Beattie et al., 2016). We compare three different representation learning methods: One-step frame prediction, contrastive predictive coding (CPC) (Oord et al., 2018), and CPC|Action, a new action-dependent variant of CPC. CPC and CPC|Action are both able to represent distributions of future observations, whereas one-step frame prediction can only represent the mean; however, frame prediction is better at paying attention to details in the observations.

Our main finding is that all three methods are able to learn a representation of the belief state of the environment. We observe that the learned representations encode important pieces of information about the environment including the agent's current position and orientation, its past trajectory, and even the position of objects in the environment. In fact, our results show that the belief representations not only encode these pieces of information, they also encode the agent's uncertainty over them—a crucial aspect of a belief state. However, we find that not all objects can be captured equally well by the learned representations, and that the representations are able to better capture those objects that have higher impact on the agent's future observations. Finally, we observe that for CPC, predicting further into the future and conditioning on actions (CPC|Action) results in the best learned belief on visually complex environments, while being more computationally efficient than the one-step frame predictor.

## 2 BACKGROUND AND NOTATION

### 2.1 PARTIALLY OBSERVABLE MARKOV DECISION PROCESSES

We consider Partially Observable Markov Decision Processes (POMDPs; Lovejoy, 1991; Cassandra, 1998) as a general framework to deal with partially-observable and stochastic environments with actions. Formally, a POMDP is a tuple $M = (\mathcal{X}, \mathcal{A}, \mathcal{O}, P, O)$ where $\mathcal{X}$ is the state space, $\mathcal{A}$ is the action space, $\mathcal{O}$ the observation space, $P$ models the dynamics and maps to each state-action couple $(x, a)$ a probability $P(y|x, a)$ over the next state $y$ and $O$ is the observation distribution that maps to each state $x$ a probability $O(\cdot|x)$ over possible observations. Typically, POMDPs also include a reward observation; however, as we are not considering the control problem here, there is no need to distinguish between the reward and the observations, so we omit the reward.

At any given time $t$, the agent acting in a POMDP has only access to some observation $o_t \in \mathcal{O}$ that gives incomplete information about the real state $x_t \in \mathcal{X}$. Thus, it has an uncertainty on the real state $x_t$ as well as on the next state $x_{t+1}$ as the dynamics depends on the state-action pair $(x_t, a_t)$. Therefore a key aspect in POMDPs is to be able to compute a belief state $b_t$, which is a probability distribution over possible states, from the current history $h_t$. More formally, at a given time $t$, the current history $h_t$ is the set of past actions and observations $h_t = \{o_0, a_0, o_1, a_1, \ldots, a_{t-1}, o_t\}$, and a belief distribution $P_b(\cdot|h_t)$ over the possible states conditioned on the history of past actions and observations. Ideally, we would like to compute the belief distribution $P_b$ or a surrogate representation $b_t \in \mathbb{R}^d$ that encodes the information with regard to $P_b$, thus capturing the uncertainty on the underlying state $x_t$.

### 2.2 CONTRASTIVE PREDICTIVE CODING

Contrastive Predictive Coding (Oord et al., 2018) (CPC) is an unsupervised representaion learning which relies on noise contrastive estimation (Gutmann & Hyvärinen, 2010; 2012) as the statistical method for learning distributions. We provide a brief overview of noise contrastive estimation approach and based on this we describe the CPC approach. Discriminating between samples coming from the data distribution (positive examples) and samples coming from another distribution (negative examples), is known as learning from comparison. A simple way to implement this general principle is via binary classification where samples coming from the data distribution will be labelled as positive examples and samples coming from another distribution will be labelled as negative examples. Then, training such a binary classifier can be a good way to learn features that encode information on the

data distribution. More precisely, assume that we have $N^+$ samples $(o_i^+)_{i=1}^{N^+}$ coming from our data distribution with probability density $\rho^+$ and $N^-$ samples $(o_i^-)_{i=1}^{N^-}$ coming from our data distribution with probability density $\rho^-$. Training a binary classifier $f$ with logistic regression consists in finding $f$ that maximises $\hat{J}(f)$:

$$\hat{J}(f) = \frac{1}{N^+} \sum_{i=1}^{N^+} \log(f(o^+)) + \frac{1}{N^-} \sum_{i=1}^{N^-} \log(1 - f(o^-)).$$

The quantity $\hat{J}(f)$ is the empirical version of $J(f)$:

$$J(f) = \mathbb{E}_{o^+ \sim \rho^+} \left[ \log(f(o^+)) \right] + \mathbb{E}_{o^- \sim \rho^-} \left[ \log(1 - f(o^-)) \right].$$

As it is shown in Goodfellow et al. (2014) the quantity $\max_f J(f)$ is simply the Jensen-Shannon divergence $D_{JS}(\rho^+|\rho^-)$ between the data distribution and the other distribution:

$$\max_f J(f) = 2D_{JS}(\rho^+|\rho^-) - \log(4).$$

In other words, by estimating this divergence, we learn how different the positive examples are from the negative examples and hope that the learned representation encodes that information.

CPC makes use of a noise contrastive estimation model to discriminate observations $o_{t+k}^+$ at a "future" time step $t+k$ from negative observations $o_{t+k}^-$, which is randomly chosen from the dataset (see Sec. 4 for details). CPC bases this estimation on a state representation $b_t$ that depends on the history up to time step $t$ and embeddings of positive and negative observations at time $t + k$. The CPC architecture takes into account the belief state representation $b_t$ by using modern memory architecture such as LSTM (Hochreiter & Schmidhuber, 1997; Xingjian et al., 2015) and GRU (Chung et al., 2014). Different possible losses based on this description can be formulated as shown in the appendix.

## 3 RELATED WORK

In this work, we are interested in learning representations that can compactly encode the belief state in partially observable problems. We also want these representations to capture information about different attributes of the state such as agent and object positions in navigation tasks. It is to be expected that compact representations of POMDPs make it easier to learn and represent models (Boutilier et al., 1999). Indeed, representations that encode important parts of the state have led to improved performance in RL tasks, both with model-based and value-based methods (Guestrin et al., 2003; Diuk et al., 2008; Boots et al., 2011; Levine et al., 2016; Higgins et al., 2017; Karkus et al., 2018).

Predictive State Representations (PSRs; Littman & Sutton, 2002) are one such expressive and compact representation, in terms of *tests* on the POMDP (Rivest & Schapire, 1993). A test is the indicator of a specific sequence of future observations given a specific sequence of actions. With an appropriate collection of tests (and their conditional distributions given histories), one can encode belief states (Rivest & Schapire, 1993; Littman & Sutton, 2002). A PSR is a collection of tests that is expressive enough to effectively encode the conditional distribution of any other test, and as a consequence any belief state in the POMDP. Recently Hefny et al. (2018) have implemented a deep variant of PSR architecture with recurrent neural networking, proving the compatibility of this idea with modern deep architectures. Our work is inspired by the idea that predicting future observations conditioned on future actions can give us an expressive state representation, and this principle guided the design of our representation learning architecture.

Sutton et al. (2011); Li et al. (2015); Jaderberg et al. (2016); Dosovitskiy & Koltun (2017); Higgins et al. (2017); Wayne et al. (2018); Igl et al. (2018) used auxiliary tasks to improve agent performance, but only partially investigated what information the learned representations encode. Dosovitskiy & Koltun (2017) uses supervised learning to learn representations that capture state information, and showed that this leads to improved performance in different ViZDoom tasks. Higgins et al. (2017) used methods that learn factored state representations (Burgess et al., 2018) and showed improved performance and effective transfer in 3D RL tasks where the agent must identify and collect good objects, while avoiding bad ones. Wayne et al. (2018) showed that the representation of their proposed agent architecture is able to capture the absolute position of the goal in a large-maze navigation task.

Schmidhuber (1991); Diuk et al. (2008); Kolter & Ng (2009); Sorg et al. (2010); Anandkumar et al. (2014) and many others prescribed or tried to estimate the state transition model of the POMDP explicitly. Although our approach learns about the dynamics of the environment, we do not directly evaluate the quality of the learned dynamics model. Instead, we focus on evaluating the quality of the learned representation, which implicitly captures the quality of the learned dynamics as well. In Section 5 we investigate the accuracy of these representations across different domains when trained with different approaches.

## 4  ARCHITECTURE AND ALGORITHM

Inspired by the PSR literature, our approach relies on predictions of future observations conditioned on future actions as a way to predict the belief state. In particular, we base our architecture on CPC, using a variant of this model to learn rich representations in virtual environments.

We now describe the architectures we use in our experiments. Figure 1 outlines the CPC|Action architecture which is a variant of CPC architecture (see Sec. 2.2). We use a GRU network (blue) to take in the history of embedded observations $z_t$ and actions $a_t$ and output the representation $b_t$ for the current time step $t$. In addition to standard CPC our architecture uses the $b_t$ to initialise an action-GRU (red) which is then fed by the future actions $\{a_{t+k}\}_{k=0}^{T-1}$. Finally, for each time step $t+k$, a multi-layer perceptron (MLP) (grey) is fed both by the output of this GRU and the positive example $z_{t+k}^+$ in order to predict 1 or by the output of the GRU and the negative example $z_{t+k}^-$ in order to predict 0 (for a more detailed description of the architecture i.e. ConvNet and fully-connected layer please see the Architecture Details section in the appendix.). We also implement the CPC without actions as exactly the same architecture as CPC|Action except that the action-GRU (red) is not fed by the future actions $\{a_{t+k}\}_{k=0}^{T-1}$ but by a dummy input $\{c_{t+k}\}_{k=0}^{T-1}$ where $c_{t+k} = c$ is a constant null vector. Finally we implement one-step frame prediction (FP; Bengio et al., 2007). This architecture learns a belief state $b_t$ for the task of predicting the next observation $o_{t+1}$ given the action $a_t$ via a transposed convolutional network (orange). Common to all 3 architectures is a convolutional neural network (CNN; LeCun et al., 1998) (yellow) that transforms the raw observation $o_t$ to a vector $z_t$. For evaluation, the belief state $b_t$ is then used by an MLP (green) in order to estimate the position, orientation or other features of the environments. It is important to note that we do not back-propagate the gradient from this MLP (green) that predicts the ground truth to the rest of the architecture.   Algorithm 1 outlines how we train the CPC|action architecture. We sample

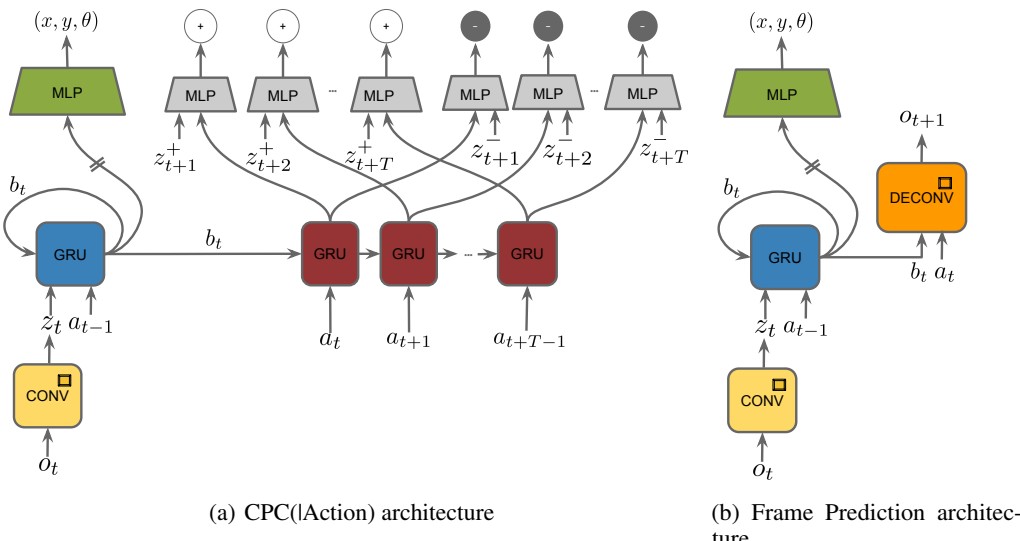

(a) CPC(|Action) architecture

(b) Frame Prediction architecture

Figure 1: Different architectures used in our experiments.

mini-batches of sub-trajectories from our dataset, and unroll the belief GRU $f$ to compute the beliefs $b_t$ for every time step. Then, for every $b_t$, we sample how far we want to predict into the future up to

a maximum of $F$. We compute the forwarded belief from the Action GRU, and then feed it to the CPC classifier with both the true future observation as the positive example, and a randomly picked observation from the mini-batch as a negative example. We average the classification losses across all time steps of the mini-batch and take a gradient step. For the frame predictor, the training procedure is similar, except we compute the prediction loss of the next future observation instead of the CPC loss for each time step. The distribution of negative examples, and the ratio of the number of positive

---

**Algorithm 1:** CPC|Action

**Data:** Belief GRU $f$, Future Prediction Length $F$, Action GRU $g$, CPC Classifier $h$

1   **for** $i \leftarrow 0$ **to** $\infty$ **do**
2     Initialise loss $\ell \leftarrow 0$ ;
3     Sample mini-batch $B$ of size $N$ from replay;
4     Compute beliefs $b_t = f(b_0, z_{1:t}, a_{1:t-1})$ ;
5     **for** *sub-trajectory* $j \leftarrow 0$ **to** $N$ **do**
6       **for** *step* $t \leftarrow 0$ **to** $T$ **do**
7         Sample $f$ uniformly from $[1, \ldots, F]$ ;
8         Let $a_{t:t+f-1}$ be the future actions starting from the current step ;
9         Let $z_{t+f}$ be the future observation $f$ steps in the future ;
10        Let $b_t$ be the belief state at current step ;
11        Sample negative example $z^-$ uniformly from $B$ ;
12        $b^a = g(b_t, a_{t:t+f-1})$ ;
13        $\ell^+ = \text{sigmoid\_cross\_entropy}(h(b^a, z_{t+f}), 1)$ ;
14        $\ell^- = \text{sigmoid\_cross\_entropy}(h(b^a, z^-), 0)$ ;
15        $\ell \leftarrow \ell + \ell^+ + \ell^-$ ;
16       **end**
17     **end**
18     $\ell \leftarrow \frac{\ell}{|B|}$ ;
19     Take gradient step to minimise $\ell$ ;
20   **end**

---

and negative examples can be an important choice for CPC and CPC|Action. We found that taking one negative observation uniformly from the rest of the mini-batch performed very well. We also found that the distribution and ratio became significantly less important when we predict further into the future.

## 5   EXPERIMENTS

In this section we describe our experimental setup and discuss the results. We would like to evaluate whether our learned representations can encode information about the underlying state of the environment from just partial observations. After motivating our results with experiments in a toy domain (Section 5.1), we present two sets of experiments in a visually rich partially observable 3D environment. In the first set of experiments (Section 5.2), we compare the three different approaches, and look into their capacity to encode the agent's position and orientation. In the second set of experiments, we delve deeper into the subject of encoding beliefs about objects' positions (Section 5.3), and the uncertainty on agent's position and orientation (Section 5.4). Additional details about the experimental setups can be found in Appendix A.

### 5.1   TOY GRIDWORLD

To give a sense of the belief representations being learned, we first present qualitative results in a toy domain, where we can see how the learned belief changes (in particular reduction in uncertainty) as the agent interacts with the environment. To evaluate the learned belief, we train a separate classifier to predict the true position and orientation of the agent from the learned belief, without letting the gradient flow back into the representation.

The toy domain is a square gridworld room. At each step the agent moves (forward or backwards) or rotates (a quarter of a circle to the left or right) at random, and it is only able to observe a square of length 5 centred on it. Fig. 2 shows screenshots from an episode with an agent trained with CPC|Action and predicting 30 steps into the future. We observe that the agent's representation encodes a belief that reflects the inherent uncertainty on the agent's position and orientation. This uncertainty is a result of partial observability and, as the agent moves around the room and observes more of the environment, it is progressively reduced, until eventually there is no more uncertainty on the agent's position and orientation for the rest of the episode. More specifically we observe that throughout the early stage of episode, Figs. 2(a) to 2(d), when the agent's observation are not informative, the agent is able to refine his belief only by ruling out the states which are not feasible under the past actions. When the agent observes the top wall at time step 32 (Fig. 2(e)) it immediately narrows down its belief to only 3 neighboring states. After that it takes the agent another 20 time steps to completely resolve the uncertainty again by using the past actions.

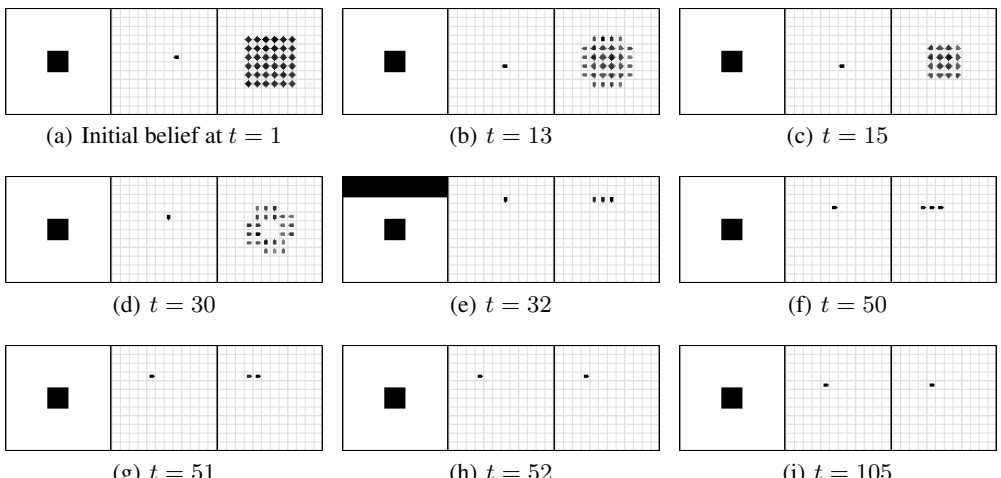

(a) Initial belief at $t = 1$     (b) $t = 13$     (c) $t = 15$

(d) $t = 30$     (e) $t = 32$     (f) $t = 50$

(g) $t = 51$     (h) $t = 52$     (i) $t = 105$

Figure 2: Frames from the agent moving at random in a gridworld (outermost cells are walls, see Fig. 2(e)). In each image, the agent's partial observation is on the left (agent and walls in black, empty spaces in white), the agent's position and orientation are on the centre, and the predicted position and orientation are on the right. The diamond-looking shapes result from flattening the beliefs for each of the four possible orientations in the same cell.

## 5.2 ALGORITHM COMPARISON

In this section we are interested in a variety of visually rich, partially observable environments, so we used four different environments of the DeepMind Lab platform (Beattie et al., 2016).

We tested whether the representations encoded the following: 1) agent's relative $(x, y)$-position and orientation $\theta$ at each time step, given the agent's initial position and orientation, 2) the agent's past relative positions and orientations up to each time step, and 3) the relative position of uncollected objects in the environment at each time step. The reason we test for the agent's past positions and orientations is because the history is necessary for the agent to remember collected objects.

We trained separate networks to estimate the learned representation and initial position and orientation and predict the associated piece of information at every time step. These networks were used only for inspection, that is, gradients did not flow through them into the representation. The positions and orientations are discretised for easier evaluation. To generate data, we used a random policy that repeats a randomly chosen action a random number of times between 1 and 5.

The four environments used in our experiment span different kinds of layouts from rooms to mazes to natural terrain, and different object positioning—fixed positions or per-episode randomised positions. Objects are collectable in all four environments. `fixed` is a single room with objects in fixed locations, `room` is a single room with objects in randomised locations, `maze` is a fixed maze with objects in randomised locations, and `terrain` is a naturalistic, hilly, terrain with desert and forest

features, and objects. In `terrain`, the map (including object positions) is randomly selected in every episode from a fixed, finite set. Fig. 3 gives examples of agent observations from each of these environments (the specific DeepMind Lab environment names are given in Appendix A).

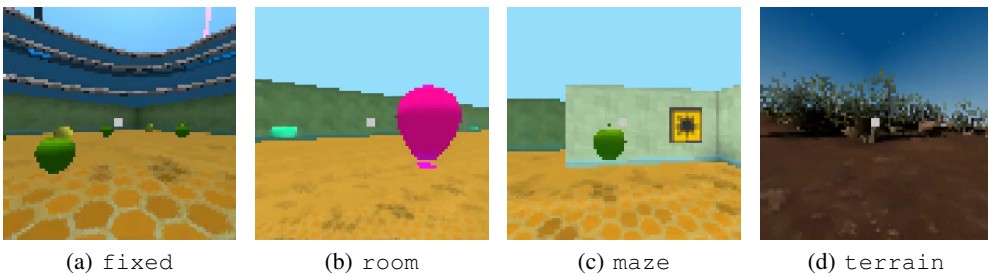

| (a) `fixed` | (b) `room` | (c) `maze` | (d) `terrain` |

Figure 3: Examples of agent observations for different environments.

We compared CPC|Action to CPC (Oord et al., 2018), as well as frame prediction (FP), which predicts the next frame given the current representation and action. For both CPC|Action and CPC approaches, we test the architectures trained from predicting 1 and 30 steps into the future. Table 1 summarises the prediction losses across all algorithms and environments[1], and we can make several observations.

| Env | Algorithm | $(x, y, \theta)$ | Past $(x, y, \theta)$ | Objects $(x, y)$ |
|---|---|---|---|---|
| `fixed` | FP | $\mathbf{0.118 \pm 0.015}$ | $0.121 \pm 0.007$ | $0.043 \pm 0.006$ |
| | CPC 1 | $0.579 \pm 0.067$ | $0.132 \pm 0.010$ | $0.049 \pm 0.005$ |
| | CPC 30 | $0.562 \pm 0.204$ | $0.118 \pm 0.010$ | $0.045 \pm 0.004$ |
| | CPC\|Action 1 | $0.689 \pm 0.057$ | $0.137 \pm 0.006$ | $0.049 \pm 0.004$ |
| | CPC\|Action 30 | $0.240 \pm 0.030$ | $\mathbf{0.100 \pm 0.007}$ | $0.040 \pm 0.003$ |
| `room` | FP | $\mathbf{0.517 \pm 0.123}$ | $0.285 \pm 0.017$ | $0.484 \pm 0.005$ |
| | CPC 1 | $2.010 \pm 0.142$ | $0.311 \pm 0.017$ | $0.498 \pm 0.008$ |
| | CPC 30 | $\mathbf{0.482 \pm 0.157}$ | $\mathbf{0.257 \pm 0.022}$ | $0.481 \pm 0.005$ |
| | CPC\|Action 1 | $2.274 \pm 0.117$ | $0.308 \pm 0.018$ | $0.484 \pm 0.005$ |
| | CPC\|Action 30 | $0.689 \pm 0.066$ | $0.276 \pm 0.029$ | $0.484 \pm 0.008$ |
| `maze` | FP | $\mathbf{0.178 \pm 0.207}$ | $0.233 \pm 0.029$ | $0.322 \pm 0.008$ |
| | CPC 1 | $0.622 \pm 0.158$ | $0.278 \pm 0.055$ | $0.330 \pm 0.009$ |
| | CPC 30 | $0.244 \pm 0.058$ | $\mathbf{0.213 \pm 0.031}$ | $0.325 \pm 0.015$ |
| | CPC\|Action 1 | $0.638 \pm 0.094$ | $0.264 \pm 0.028$ | $0.323 \pm 0.010$ |
| | CPC\|Action 30 | $\mathbf{0.182 \pm 0.034}$ | $\mathbf{0.206 \pm 0.029}$ | $0.323 \pm 0.010$ |
| `terrain` | FP | $1.831 \pm 0.162$ | $0.405 \pm 0.077$ | $\mathbf{0.181 \pm 0.084}$ |
| | CPC 1 | $3.393 \pm 0.252$ | $0.417 \pm 0.074$ | $0.307 \pm 0.174$ |
| | CPC 30 | $2.280 \pm 0.853$ | $\mathbf{0.340 \pm 0.104}$ | $\mathbf{0.131 \pm 0.185}$ |
| | CPC\|Action 1 | $3.348 \pm 0.482$ | $0.414 \pm 0.042$ | $0.312 \pm 0.049$ |
| | CPC\|Action 30 | $\mathbf{1.589 \pm 0.358}$ | $0.344 \pm 0.065$ | $\mathbf{0.139 \pm 0.136}$ |

Table 1: Evaluation classification losses after $2 \cdot 10^5$ mini-batch updates for the 5 algorithm settings across all 4 environments over 2 seeds.

First, predicting 30 steps into the future with CPC and CPC|Action significantly outperforms predicting only 1 step in the future. The poor performance of CPC 1 and CPC|Action 1 is evidence that using a contrastive loss allows the representation to ignore more details of observations compared to FP. Predicting 30 steps into the future with the CPC methods allows representations to encode at least as much relevant information as FP, and at a lower computational cost. It is possible to formulate a version of FP that also predicts further into the future at an even greater computational cost, but it is not clear how much that can improve the learned belief since FP cannot represent distributions over observations, only the mean.

---

[1]The losses were computed using generated episodes not used for training.

Second, in environments with simple observations (not `terrain`), all three approaches (FP, CPC 30, CPC|Action 30) are able to accurately encode the agent's position and orientation, and perform reasonably well encoding the agent's past position and orientation. FP consistently edges out the others in encoding position and orientation. An inspection of the predictions from videos of the evaluation episodes confirm this result. However in `terrain` with more complex observations, we see from Table 1 that the prediction task is more challenging for all approaches, and there is a larger gap between the accuracy of the predictions. In Figs. 4(a) to 4(c), we see typical examples where all algorithms can accurately predict the position and orientation of the agent. In Figs. 4(d) to 4(f), we see typical examples of prediction mistakes corresponding to each of the approaches. In particular, mistakes from FP are noticeably worse than CPC and CPC|Action 30, and CPC|Action performs the best.

Third, for past position and orientation, the general trend is that both CPC approaches are slightly better than FP. This is seen in Table 1 and in evaluation videos. The CPC methods (Figs. 4(b), 4(c), 4(e) and 4(f)) are noticeably better than FP (Figs. 4(a) and 4(d)).

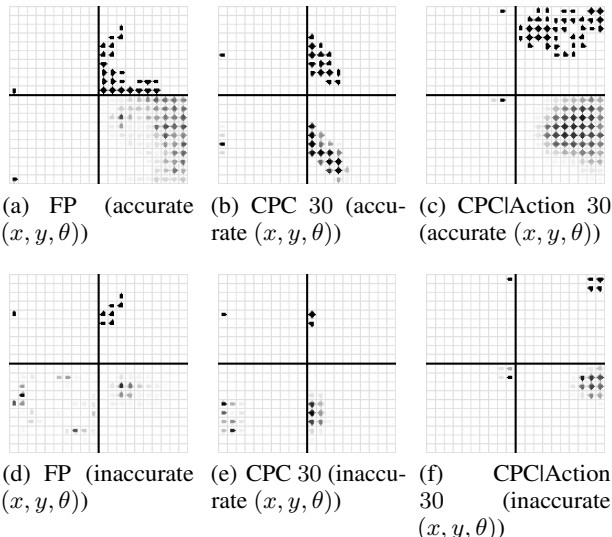

(a) FP (accurate $(x, y, \theta)$)

(b) CPC 30 (accurate $(x, y, \theta)$)

(c) CPC|Action 30 (accurate $(x, y, \theta)$)

(d) FP (inaccurate $(x, y, \theta)$)

(e) CPC 30 (inaccurate $(x, y, \theta)$)

(f) CPC|Action 30 (inaccurate $(x, y, \theta)$)

Figure 4: Example predictions for `terrain`. In each image, ground truths are on the top, predictions on the bottom, $(x, y, \theta)$ (ground truth and predictions) on the left, and past $(x, y, \theta)$ on the right. Figs. 4(a) to 4(c) show examples of accurate $(x, y, \theta)$ predictions, Figs. 4(d) to 4(f) show examples for inaccurate $(x, y, \theta)$ predictions.

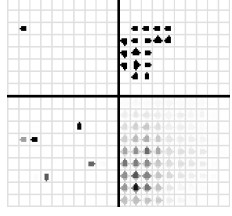

Figure 5: Symmetry of the $(x, y, \theta)$ prediction with the frame predictor in `room`. Ground truths are on the top, predictions on the bottom, $(x, y, \theta)$ (ground truth and predictions) on the left, and past $(x, y, \theta)$ on the right.

Fourth, we take a closer look at `room` as it is an interesting environment because it is almost symmetric in both $x$ and $y$ axes. The room is almost a square, measuring 9 by 10 units. There are faint vertical pulses of light on the walls that move from small to larger $(x, y)$, and they are the only symmetry-breaking elements. Table 1 and video inspection suggest that the representations trained with the three algorithms do not allow asymmetries to be consistently resolved. Fig. 5 shows an example of the symmetry in position predictions, reflecting uncertainty about the position and orientation. The higher position and orientation prediction errors of FP and CPC|Action 30 in `room` are mainly due to the ambiguity from symmetry. We are unsure why CPC is slightly better than CPC|Action at paying attention to the moving vertical lines of light on the walls, but we speculate it is because knowledge of actions taken does not help break the symmetry, and because CPC|Action may be encoding the action-dependent dynamics.

Finally, for object positions, we see varied results in Table 1, depending on the type of environment. For `fixed`, unsurprisingly, all predictors have similar accuracy, since objects are fixed. It is likely that the information is not encoded in the representation—as our results in Section 5.3 suggest—but in the evaluator which simply memorises the fixed positions. For `room` and `maze`, which randomise object locations each episode, the prediction errors indicate that the representation is unable to encode

any information object position. Finally, for `terrain`, which has a finite set of fixed object positions, the three approaches FP, CPC 30 and CPC|Action 30 allow for reasonably accurate predictions. We believe that the representations only encode information about the specific map instance, from which the evaluators can decode object position. In Section 5.3 we discuss the issue of representing objects in more detail.

## 5.3 INCREASED OBJECT INTERACTION

Table 1 shows that none of the approaches are able to encode much information about object positions (cf. `room` and `maze`). We hypothesise that the objects are not a significant enough part of the observations to warrant the CPC algorithms to pay attention to them, and there is no reason for FP to continue to remember objects once they go out of the agent's view.

To test this hypothesis, we constructed two simple DeepMind Lab environments: `non teleport`, where objects cannot be interacted with, and `teleport`, where objects, when touched, teleport the agent back to its initial position. Both environments are a small square room with the agent's initial position in a notch on the wall (to create asymmetry), and two visually different objects are placed in random positions at each episode.

The teleporting interaction results in a drastic change in the observations of the agent, which should force the representations to encode information about these objects in order to better predict future observations. In contrast, non-interactive objects (in `non teleport`) are equally visible to the agent, but do not cause drastic changes to observations.

Table 2 shows the losses for evaluating the prediction of object position for `teleport` and `non teleport`. All of the algorithms are significantly better at encoding information about the position of the objects with the teleport interaction, with CPC|Action 30 being slightly better than the others. Fig. 6 shows screenshots of an evaluation video, where we see that in the case of `non teleport` (Figs. 6(a) and 6(b)) the representation is only able to react to immediately visible objects. As soon as the agent turns away, the representation no longer contains the object information. However, in the case of `teleport` (Figs. 6(c) and 6(d)), we see that the representations are able to remember information about the objects even after the agent has turned away and moved elsewhere.

| Algorithm | Objects $(x, y)$ | |
| --- | --- | --- |
| | non teleport | teleport |
| FP | $0.148 \pm 0.003$ | $0.108 \pm 0.014$ |
| CPC 30 | $0.168 \pm 0.001$ | $0.137 \pm 0.017$ |
| CPC|Action 30 | $0.164 \pm 0.002$ | $\mathbf{0.086 \pm 0.020}$ |

Table 2: Evaluation classification losses on object position.

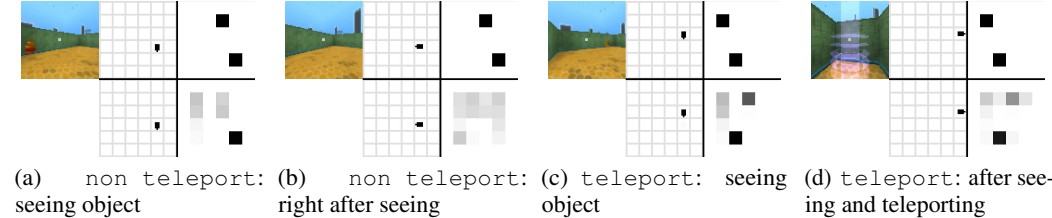

(a) `non teleport`: seeing object

(b) `non teleport`: right after seeing

(c) `teleport`: seeing object

(d) `teleport`: after seeing and teleporting

Figure 6: Example predictions for `non teleport` and `teleport`. Ground truths are on the top, predictions on the bottom, $(x, y, \theta)$ (ground truth and predictions) on the centre, object $(x, y)$ on the right, and frame seen by the agent on the left.

Furthermore, in Figs. 6(c) and 6(d) we see that the representation is able to maintain uncertainty over the object positions. When the agent only sees one of the two objects, the position of the second object is still uncertain, but the representation is already able to encode some negative evidence, narrowing down the possible locations of the second object.

## 5.4 RICHER UNCERTAINTY OVER POSITION

In the DeepMind Lab environments, there is not an instance of uncertainty over the agent's position and orientation similar to the toy Gridworld (Section 5.1) due to being able to see far into the distance in a first-person view. Therefore we constructed a simple DeepMind Lab environment with two parallel hallways (see Fig. 7(a)) to demonstrate this kind of uncertainty in a 3D environment. The agent randomly starts in one of the two hallways, and its position can only be resolved near the exit of the hallways (we do not give the agent's initial position to the evaluator). Fig. 7(b) illustrates (for CPC|Action 30) how, initially, the representation cannot distinguish in which hallway the agent is. Fig. 7(b) illustrates the representation immediately resolving the location when the agent able to peek out. This behaviour is consistent for all three algorithms: FP, CPC 30 and CPC|Action 30. Thus, even in 3D environments, we are able to learn a belief that can encode a richer uncertainty over the agent's position.

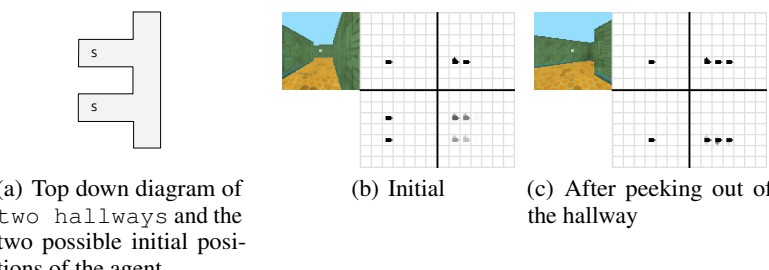

(a) Top down diagram of `two hallways` and the two possible initial positions of the agent.

(b) Initial

(c) After peeking out of the hallway

Figure 7: Example predictions for `two hallways`. Ground truths are on the top, predictions on the bottom, $(x, y, \theta)$ (ground truth and predictions) on the centre, past $(x, y, \theta)$ on the right, and frame seen by the agent on the left.

## 6 CONCLUSION AND FUTURE WORK

Using a glass box approach, we investigated the quality of representations learned by three different methods: FP, CPC, and CPC|Action. Specifically, we considered a variety of first-person 3D navigation environments, and looked at whether the representation can encode a belief on different aspects of the environment. We found that FP, CPC 30 and CPC|Action 30 are all able to learn representations that encode the agent's position and orientation, the agent's trajectory (previous positions and orientations)—cf. Table 1. The position of objects can be encoded as well, provided that interacting with the objects strongly impacts the agent's future observations (Table 2).

More importantly, the representations also encode the agent's uncertainty over its position and object positions. We showed that this uncertainty is reduced as the agent obtains more information from the environment, including negative evidence, e.g., when the agent sees where the object is not (Fig. 6).

In visually simple environments (e.g. `fixed`), FP was the best at encoding agent position and orientation. In visually complex environment (`terrain`), CPC 30 and CPC|Action 30 performed best, with multi-step predictions being the key to their success, and action-conditioning providing further improvements.

There remains much interesting future work to pursue. We believe the ability of these representations to learn various belief concepts can be further explored to improve performance and generalisation in multi-task settings, by transferring concepts across tasks. The capability of encoding uncertainty can also be useful for learning policies that efficiently explore partially observable environments by acting to reduce uncertainty on the agent's belief (as in Bayes-optimal exploration; Wilson et al., 2007; Kolter & Ng, 2009; Sorg et al., 2010; Asmuth & Littman, 2012; Ghavamzadeh et al., 2015). As another direction, the three methods we considered can go beyond predicting only visual observations to other modalities of sensory inputs, such as proprioception and touch sensors (Amos et al., 2018). This should lead to belief representations that encode a richer variety of information about the environment and its structure.

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

## A    IMPLEMENTATION DETAILS

**Environments.**    Table 3 gives the names of the four environments (levels) of the DeepMind Lab platform (Beattie et al., 2016) that we used.  The latter three tasks are custom DeepMind Lab environments that we created.

| Name in this paper | `dmlab30` name |
|---|---|
| `fixed` | `seekavoid_arena_01` |
| `room` | `rooms_collect_good_objects_train` |
| `maze` | `nav_maze_random_goal_01` |
| `terrain` | Smaller variant of `natlab_fixed_large_map` |
| `teleport` | — |
| `non teleport` | — |
| `two hallways` | — |

Table 3: Correspondences of our environments to DeepMind Lab levels.

**Frame Reconstruction Loss.**    For frame prediction, we normalise the pixel colour values to be between $0$ and $1$ and use the sigmoid cross-entropy loss.

**CPC Losses.**    We implement the CPC losses differently from Oord et al. (2018). They score examples for the contrastive loss at $k$ time steps in the future with $f_k(o) \doteq \exp(\text{conv}(o)^\top W_k b_t)$ for a matrix $W_k$, where $b_t$ is the current belief. The contrastive loss to be minimised at time step $t$ and looking $k$ steps in the future is

$$-\ln \frac{f_k(o^+)}{f_k(o^+) + \sum_{j=1}^m f_k(o_j^-)},$$

where the positive example is $o^+ = o_{t+k}$ (the observation at time step $t + k$), and the negative examples are $(o_1^-, \ldots, o_m^-)$, which can be sampled from a minibatch of data.

In our case, the score function $f$ is a one-hidden-layer perceptron with ReLU activation in the hidden layers, taking as inputs the concatenation of $\text{conv}(o)$ and $b_t$. The contrastive loss to be minimised at time step $t$ and looking $k$ steps in the future is the binary classification loss

$$\sigma(f(o^+, b_t)) + \sigma(-f(o^-, b_t)),$$

where $\sigma$ is the sigmoid function, the positive example is $o^+ = o_{t+k}$ (the observation at time step $t + k$), and the negative example $o^-$ is drawn uniformly at random from the minibatch (including different time steps of the trajectory $o_{t+k}$ belongs to).

**Architecture Details.**    A diagram of the architecture is in Fig. 1.

The observations from DeepMind Lab are $84 \times 84$ pixels, each pixel consisting of three bytes representing RGB values respectively. This observation is passed through our convolutional network, which has three convolutional layers with filter sizes $8, 4, 3$, strides $4, 2, 1$, and number of filters $32, 64, 64$ respectively, and a final hidden layer of size $512$ with ReLU activations after every layer.

The belief GRU takes as input the concatenation of $z_t$ and $a_{t-1}$ and outputs $b_t$, where $z_t$ is the output of size $512$ of the conv net after passing in the observation $o_t$, and $a_{t-1}$ is a one-hot vector of the discrete action. The belief GRU has a hidden size of $512$ and thus the output $b_t$ also has a size of $512$.

The action GRU also has a hidden size of $512$, and takes $b_t$ as the initial hidden state. It takes one-hot vectors of the discrete actions $a_t$ as input, and outputs a forwarded belief of size $512$.

The forwarded belief is then concatenated with the corresponding positive example $z^+$, which is the output of size $512$ of the same convnet as before of a positive observation example $o^+$. The concatenation is then fed to the contrastive discriminator which is an MLP with a hidden layer of size $512$ and ReLU activations, and a linear output of size $1$. This output of size $1$ is then fed into a sigmoid cross-entropy loss for classifying the positive example as class $1$. A similar process is used for classifying a negative example that uses the same forwarded belief and contrastive discriminator.

For the frame predictor, the deconv network architecture is the transpose of the same convnet we use for observations.

For evaluating the encoded information the belief $b_t$ of position and orientation, past position and orientation, and object positions, we use a two hidden layer MLP with each hidden layer having size $512$ and ReLU activations. The input to the MLP is the concatenation of $b_t$. In the case of `fixed`, `room`, `maze` and `terrain` levels, we provide the one-hot of the agent's initial discretised position and orientation to break the symmetry in these environments. The output is a softmax for predicting position and orientation, a grid of sigmoids for past position and orientation, and again a grid of sigmoids for object positions. We discretised the orientation into the 4 cardinal directions. For position, we did the following discretisations: `fixed` is $9 \times 10$, `room` is $9 \times 10$, `maze` is $10 \times 5$, `terrain` is $10 \times 10$, `two hallways` is $7 \times 7$, and `teleport` and `non teleport` are $6 \times 6$.

**Training Details.** To gather data, we used a policy that picks an action at random, and then repeats that action between $1$ and $5$ times. We trained using a distributed framework, where we used $128$ processes to interact with the environment and push trajectories into a FIFO replay buffer. The trajectories are partitioned into 100-step sub-trajectories, and the replay buffer has a max capacity of $5 \cdot 10^4$ sub-trajectories. We have one training process that samples mini-batches of $64$ sub-trajectories uniformly from the replay buffer and trains using the Adam optimiser (Kingma & Ba, 2015) in TensorFlow (Abadi et al., 2016) with default hyperparameters with a learning rate of $0.0005$. The belief GRU is shared with the $128$ interacting processes, as they also push the hidden state of the GRU of the first step of each sub-trajectory into the replay buffer as well. In the training process, after sampling a mini-batch, we use this stored initial hidden state and unroll the belief GRU for the rest of the steps to compute the beliefs.

