# OpenReview forum: "Neural Predictive Belief Representations"
_ICLR.cc/2019/Conference_

### Official Review · AnonReviewer1 · 2018-10-31
**Interesting ideas, excellent experiments but a big question mark and significance hard to assess**

**Rating:** 5
**Confidence:** 3

**Review:**

This paper learns a deep model encoding a representation of the state in a POMDP using one-step frame prediction or a  contrastive predictive coding loss function. They evaluate the learned representation and shows it can be used to construct a belief of the state of the agent.

Using deep networks in POMDP is not new, as the authors pointed out in their related work section. Thus I believe the originality of the paper lies in the type of loss used and the evaluation of the learned representation trough the construction of a belief over current and previous states. I think this method to evaluate the hidden state has the potential to be useful should one wish to evaluate the quality of the hidden representation by itself. In addition, I found the experimental evaluation of this method to be rather extensive and well conducted, as the authors experimented on 3 different (toy) environments and use it to quantify and discuss the performance of the three model architecture they develop.

On the other hand, the authors mention that other works have already shown that learned representations can improve agent performance, can be learned by supervised learning (= predicting future observations) or can be useful for transfer learning. So in that context, I am not sure the contributions of this paper are highly significant as they are presented. To better highlight the strength the evaluation method, I think it could be interesting to check that the accuracy in belief prediction is correlated with an improvement in agent performance or in transfer learning tasks. To better highlight the interest of the CPC loss, I think it could be interesting to compare it to similar approaches, for example the one by Dosovitskiy & Koltun (2017).

I found the paper reasonably clear. However, the following sentence in the appendix puzzled me. "The input to the [evaluation] MLP is the concatenation of b_t and a one-hot of the agent’s initial discretised position and orientation." I may have missed something, but I do not understand how the model can contain the uncertainty shown in the experimental results if the agent's initial position is provided to the current or past position predictor.

---

> ### Author Response · Authors · 2018-11-26
> **Thanks for the review.**
>
> We would like to thank the reviewer for their review and comments.
>
> 1. "The authors mention that other works have already shown that learned representations can improve agent performance."
>
> For this work, we have chosen to understand neural belief representations. In this sense, the problem we are studying is representation learning, and whether the neural representations we propose to learn are effective belief representations. To motivate why one should care about learning effective belief representations, we presented examples of previous work where better representations led to improved performance in different tasks. However, one cannot say that these works have used predictive belief representations, and the degree to which each studies the learned representations varies.
>
> 2. "I think it could be interesting to check that the accuracy in belief prediction is correlated with an improvement in agent performance or in transfer learning tasks."
>
>   There are three different problems being conflated here. First, solving a specific task. For this, task performance is the clear success criterion. Second, using neural belief representations for solving a specific task. Here, too, task performance is the criterion of interest, but it is only indirectly informative about how good the neural belief representation is, as a belief representation. Third, learning strong neural belief representations. Here, looking at the quality of the learned beliefs is a more sensible criterion than performance in any tasks, although looking at performance across tasks could be reassuring.
>
> 3. If the initial position is given to the MLP, how can there be any uncertainty?
>
> We have made an imprecise statement in that regard, which we will fix: Only the discretized initial position and orientation was being fed, and the agent may require several actions to move from one square in the grid to another, or to change its discretized orientation. There are some additional comments to be made about the reviewer's remark.
>
> The agent's initial discretized position and orientation were only fed to the MLP (alongside the RNN output) for the levels 'fixed', 'room', 'maze', and 'terrain'. Thus the uncertainty on the agent's position in 'two hallways' is natural (and for 'teleport' and 'non-teleport' it can be easily eliminated because the environment is fixed).
>
> We can speculate some additional reasons for the uncertainty in 'fixed', 'room', 'maze', and 'terrain'. In principle, with the full sequence of actions as well as the initial position, on a fixed environment, the agent's position and orientation could be maintained. This kind of simulation is unlikely to be happening though, as we can see from Figures 4 and 5, by looking at the history predictions: The MLP cannot decode the full history out of the representation.
>
> Moreover, in the case of 'terrain' , the terrain affects the agent's position, so the simulation could not be used for accurate position/orientation prediction. (This is also aggravated by the fact that 'terrain' is not fixed, but the terrain is randomly drawn at each episode from a fixed set of instances.)

---

### Official Review · AnonReviewer3 · 2018-11-04
**Good paper, some details missing or unclear**

**Rating:** 7
**Confidence:** 4

**Review:**

** Summary **
The authors evaluate three different representation learning algorithms for partially observable environments. In particular, they investigate how well the learned representation encodes the true belief distribution, including its uncertainty.
They propose an extension to a previous algorithm and evaluate all three algorithms on a range of tasks.

** Clarity **
The paper is well written and overall easy to follow.

** Quality **
The paper evaluates the described algorithms on a sufficiently large set of tasks. There is no theoretical analysis.

** Originality & Significance **
While the authors propose a novel extension to an existing algorithm, I believe the value of this work lies in the detailed empirical analysis.

** Missing Citations **

I believe two recent papers (this year's ICML) should be mentioned in the related work section as they propose two representation learning algorithms for POMDPs that, as far as I can tell, are not yet mentioned in the paper but quite relevant to the discussed topic. [1] Because it also uses PSRs and [2] because it explicitly learns a belief state. It would be interesting to see how [2] compares in terms of performance to FP and CPC(|Action).

[1] Hefny, A., Marinho, Z., Sun, W., Srinivasa, S. & Gordon, G.. (2018). Recurrent Predictive State Policy Networks. Proceedings of the 35th International Conference on Machine Learning, in PMLR 80:1949-1958

[2] Igl, M., Zintgraf, L., Le, T.A., Wood, F. & Whiteson, S.. (2018). Deep Variational Reinforcement Learning for POMDPs. Proceedings of the 35th International Conference on Machine Learning, in PMLR 80:2117-2126

** Question **

I have several questions where I'm not sure I understand the paper correctly:

1.) Why can FP only predict the mean? For example, one could use a PixelCNN as decoder, which would allow to learn an entire distribution, not just the mean over images.
2.) The problem that CPC and CPC|Action is unable to predict objects if they don't influence the future trajectory doesn't seem surprising to me because whether an image is a positive or negative example can usually be determined by the background, the object is not necessary to do so. In other words, this is a problem of how the negative samples are chosen: If they were constructed using a simulator that shows the same background but without the objects, the belief would need to start encoding the presence of objects. Is this correct or am I missing something?
3.) Am I correct in thinking that CPC(|Action) would not be applicable to properly estimate the belief distribution in the presence of noise, i.e. for example when estimation the exact location based on sensors with Gaussian noise?

** Overall **

* Pros:
- Extensive, interesting evaluation
- Novel CPC|Action algorithm

* Cons:
- No theoretical analysis/justification for claims
- There are several subtleties that I am not sure are sufficiently discussed in the paper (see my questions above)

---

> ### Author Response · Authors · 2018-11-26
> **Thanks for the review.**
>
> We would like to thank the reviewer for their review and comments. The reviewer has positive comments concerning the empirical evaluation and the new algorithm presented. Its major concern is with respect to the lack of theoretical analysis and raises also some minor concerns regarding the different architectures. We will try to address those comments in the remaining.
>
> Concerning the lack of theoretical analysis: This works is an empirical evaluation via a glass-box approach of three different algorithms that shape the representation of a belief state. This approach is new to our knowledge. The evaluation clearly illustrates that the representations learnt with the different architectures are able to encode information related to the belief state of the agent such as its position and orientation.
>
> Concerning the missing citations: We agree with the reviewer. We will add those citations and a discussion in the related work section. The main difference between our work and those works is that we are not trying to learn an explicit belief probability vector, and we go more in-depth with our glass-box evaluation in quantifying what sorts of information is being learned and why.
>
> Concerning the questions : We agree with the reviewers on the 2 first points. You could indeed learn a distribution for the frame prediction and encoding the objects in the representation of the belief need a more clever choice of negative examples.  To expand on that, the choice of negative examples is critical to what information is encoded, and indeed we can add more prior knowledge in shaping this choice; but there is always a trade-off in how much we can affect the design. Using a simulator that can remove objects from the background is only feasible when one has complete control over the environment, and so is very task-specific. We are more generally interested in broader approaches.
>
> For the third point, CPC(|Action) should have no problem with sensor noise, as long as the noise is not overwhelming to the point where positive and negative examples are indistinguishable. The result will be simply that they learn a noisier estimation of all encoded information.

---

### Official Review · AnonReviewer2 · 2018-11-05
**Review for "Neural Belief Representations"**

**Rating:** 4
**Confidence:** 3

**Review:**

# Review for "Neural Belief Representations"



The authors argue in the favor of belief representations for partial observable Markov decision processes. The central argument is that uncertainty needs to be represented to make optimal decision making. For that aim, three belief representations based on sufficient statistics of the future are evaluated and compared in a set of disective studies. Those studies find that predicting the future results in uncertainty being represented in the state representations, although they differ in quality.

I found the paper hard to follow for various reasons.

- NCE is reviewd, while CPC is not. I would have found a review of CPC as well to help my understanding, especially to draw the line between CPC and CPC|Action.
- In 2.1., $b_t$ is defined as a probability, while it is the output of a neural network later. This is formally incompatible, and I found  the connection not well explained. From my understanding, $b_t$ is a vector that represents the sufficient statistics if learning works. The probability interpretation is thus stretched.
- The architecture description (starting from the second paragraph on page 4) feels cluttered. It was clearly written as a caption to Figure 1 and hence should be placed as such. Still, stand alone texts are important and in my humble opinion should be augmented with equations instead of drawings. While the latter can help understanding, it lacks precision and makes reproduction hard.
- The MLP to predict the ground truth is not sufficiently described in the main text. I think it needs to go there, as it is quite central to the evaluation.

Since the manuscript is half a page under the limit, such improvements would have been feasible.

Apart from the quality of the manuscipt, I like the fact that a disective study was done in such a way.

However, I would have liked to see more comparisons, e.g. in $(x, y, \theta)$ environments it is also possible to obtain quite good approximations of the true posterior via particle filtering. Also, other more straightforward approaches such as MDN-RNNs can represent multiple maxima in the probability landscape; this would have enabled to examine the benefit of conditioning on actions in a different context.

Right now, it is unclear what the paper is about. On the one hand, it does a focused disective study with well controlled experiments, which would be a good fit if many different models were considered. On the other hand, it advertsises CPC|Action; but then it falls short in evaluating the method in more challenging environments.

To sum it up, I feel that the paper needs to be clearer in writing and in experimental structure. The currently tested hypothesis, "does CPC|Action perform better than CPC and FP in a set of well controlled toy environments" is, imho, not of broad enough interest.

---

> ### Author Response · Authors · 2018-11-26
> **Thanks for the review.**
>
> Thank you for reading and commenting on our work. We address the following main concerns.
>
> The reviewer has raised the issue that the paper is hard to follow and recommends a list of changes to improve the quality of the manuscript.  In the revised manuscript we will have addressed these issues. (Items 1-4 the change list in general response.)
>
> 1. Little reviewing of CPC, CPC|Action: We will add a more detailed review to address this issue.
>
> 2. The probability interpretation for b_t is stretched: We have been imprecise here and we will fix it. Indeed, b_t is the output of a recurrent neural network. We want to clarify that the output of the neural network is itself not a probability vector, but just the output of a neural network; however the information contained in this output is quite rich, and our experiments show how this output is encoding various pieces of information about the history, approximating an encoded sufficient statistic of the history.
>
> 3. The architecture description should be precise: We will fix the presentation accordingly.
>
> 4. The MLP to predict the ground truth is not sufficiently described in the main text: We present it as an implementation detail in the appendix, but if after all the other changes there is space left, we will move it to the main text. In any case we will make sure that the main text clearly points to the implementation details.
>
> 5. The reviewer has also raised the concern that our results only  provide evidence for the hypothesis “that CPC|Action performs better than CPC and FP in a set of well controlled toy environments' and  ‘it falls short from evaluating in more challenging environments’.
>
>     Our goal in this work is to investigate the effectiveness of various unsupervised training schemes to learn belief representations. This paper fits in the literature as a "proof-of-concept" work where we do something non-trivial that has not been done before, and we take a careful look at we have developed to give the community a good understanding of it.
>
>     Hence, our main purpose was not to find out whether CPC|Action would perform better than CPC/FP, but whether it would be at all possible to learn representations that can encode meaningful beliefs about the environment state. Part of understanding these belief representations requires measuring performance and comparing different approaches (CPC|Action, CPC and FP in our case), but the message is broader than the outcome of testing whether one method performs better than another one.
>
>     Our contributions are of broad interest because the task of learning belief representations using unsupervised prediction tasks is more challenging and less demanding than doing so with supervision. It is worth pointing out that there is no evidence that learn belief representations with supervision on the state is easy, so what we have demonstrated was by no means obvious or easy.
>
> 6. More comparisons, e.g., particle filtering and MDN-RNNs.
>
>     While we recognize that adding these comparisons would strengthen the experimental results, we believe that their absence is not detrimental to the contributions of the paper, taking our goal into consideration. There are some remarks to be made about each of the suggested approaches, and we will add these to the main paper, so that we clarify the role of the experimental study in the paper and in the literature.
>
>     Particle filtering would presumably give better beliefs, but for a price: It would require more human intervention in the training than the kind of learning regime we are interested in. We wish learning to be as close to end-to-end as possible. Nevertheless, the performance of PF would be a good "ceiling line" for the performance of the different end-to-end approaches.
>
>     We see Mixture Density Networks being used in three places: In the probes (belief decoders), in the CPC variants, or in the frame prediction. For the first two possibilities, we have already represented the distribution explicitly, which sidesteps the need for networks that can encode probabilities. For the frame prediction, especially multistep, MDNs and other networks (e.g., VAEs or IQNs) might make a significant difference, especially if we were to consider frame prediction with action-conditioning. We are considering exploring these in future work, to know how far CPC|Action can go, and whether it can be used to train representations that are as rich as those one could conceivably train with networks that predict distributions.

---

### Author Response · Authors · 2018-11-26
**General comments and change list.**

We would like to thank the reviewers for their comments. We address those comments by the following changes in the paper. In addition, we provide a specific reply for each reviewer.

1.We have consolidated our review of CPC in Sec. 2.2. This addresses a concern raised by AnonReviewer2.

2. We have corrected the definition of b_t---it is the output of a recurrent neural network. What we show experimentally is that b_t contains enough information to parameterize accurate beliefs over state variables. This addresses a concern raised by AnonReviewer2.

3. We adjusted the description of the architecture so that it is clear and precise by moving the algorithm description to the main text and adding more description, as much as space allowed us. This addresses a concern raised by AnonReviewer2.

4. We referred to a more detailed description of the ground-truth MLP predictor in the main text. More implementation details can be found in the appendix. This addresses a concern raised by AnonReviewer2.

5. We have corrected the statement about the inputs of the ground truth MLP in the appendix. Only the discretized initial position and orientation was being fed to the MLP, and only for the levels 'fixed', 'room', 'maze', and 'terrain'. This addresses a concern raised by AnonReviewer1. We have also changed the main text to answer the reviewer's question (cf. our response to AnonReviewer1).

6. We added the citations requested by AnonReviewer3 in the related work section.

---

### Meta-Review · Area_Chair1 · 2018-12-13
**Concerns about the experiments and paper clarity.**

**Confidence:** 4
**Recommendation:** Reject

**Metareview:**

This paper proposed an unsupervised learning algorithm for predictive modeling. The key idea of using NCE/CPC for predictive modeling is interesting. However, major concerns were raised by reviewers on the experimental design/empirical comparisons and paper writing.  Overall, this paper cannot be published in its current form, but I think it may be dramatically improved for a future publication.